# Recognition of Blocking Categories for UWB Positioning in Complex Indoor Environment

**DOI:** 10.3390/s20154178

**Published:** 2020-07-28

**Authors:** Yaguang Kong, Chuang Li, Zhangping Chen, Xiaodong Zhao

**Affiliations:** School of Automation, Hangzhou Dianzi University, Hangzhou 310018, China; ygkong@hdu.edu.cn (Y.K.); czp.apple@hdu.edu.cn (Z.C.); xdzhao@hdu.edu.cn (X.Z.)

**Keywords:** UWB indoor positioning, NLOS status recognition, CIR signal characteristics, mRMR feature selection, DT-SVM

## Abstract

The recognition of non-line-of-sight (NLOS) state is a prerequisite for alleviating NLOS errors and is crucial to ensure the accuracy of positioning. Recent studies only identify the line-of-sight (LOS) state and the NLOS state, but ignore the contribution of occlusion categories to spatial information perception. This paper proposes a bidirectional search algorithm based on maximum correlation, minimum redundancy, and minimum computational cost (BS-mRMRMC). The optimal channel impulse response (CIR) feature set, which can identify NLOS and LOS states well, as well as the blocking categories, are determined by setting the constraint thresholds of both the maximum evaluation index, and the computational cost. The identification of blocking categories provides more effective information for the indoor space perception of ultra-wide band (UWB). Based on the vector projection method, the hierarchical structure of decision tree support vector machine (DT-SVM) is designed to verify the recognition accuracy of each category. Experiments show that the proposed algorithm has an average recognition accuracy of 96.7% for each occlusion category, which is better than those of the other three algorithms based on the same number of CIR signal characteristics of UWB.

## 1. Introduction

### 1.1. Research Status

Outdoor positioning technologies such as the Global Positioning System (GPS) can achieve good results in outdoor positioning, but they cannot achieve good positioning results in indoor positioning. Note that indoor position perception plays an important role in many applications such as tracking. Thus it is important and necessary to do more investigation and exploration on indoor positioning technologies. Among the existing indoor-positioning systems, ultra-wide band (UWB) technology has become one of the most promising methods due to its high precision, good time-delay resolution, low power consumption, and high robustness in complex indoor environments. In particular, it has significant advantages over other indoor positioning technologies in terms of accuracy [1]. The UWB positioning system shows the best performance under line-of-sight (LOS) propagation conditions, and the positioning accuracy can reach centimeter level. However, due to the complexity of the indoor environment, there are often barriers between fixed base station (FS) and mobile base station (MS), namely non-line-of-sight propagation (NLOS) conditions, which will lead to the blocking of the signal propagation path between FS and MS and introduce positive deviation [2]. The propagation of UWB under NLOS will cause the measured distance between FS and MS to be greater than the actual distance, then result in a sharp decline in the final positioning accuracy. Therefore, the status recognition of NLOS in the UWB indoor positioning system is crucial to improving the performance of the UWB-based positioning system, thus it has become one of the research hotspots of many scholars [3,4,5,6].

Note that NLOS state recognition algorithms based on distance estimation [7,8,9] and channel impulse response (CIR) features of UWB [10,11,12,13,14,15,16] are most commonly used to identify NLOS propagation state. However, compared with the CIR feature method, the distance estimation will cause extra delay due to the collection and calculation of distance. Moreover, the real-time performance of these two methods is often unsatisfactory, and the detection accuracy often fails in meeting actual demands. Furthermore, six features (energy, maximum amplitude, rise time, average excess delay, root mean square of delayed spread, and kurtosis) were extracted by collecting a large amount of experimental data under LOS and NLOS conditions and analyzing the CIR waveforms [10]. Least square support vector machine (LS-SVM) was utilised to classify LOS and NLOS states, but the relevance of features was not considered, so the recognition accuracy was limited. A method was proposed to identify LOS and NLOS states by identifying the signal characteristics under LOS/NLOS using the maximum likelihood ratio [11,12]. Kurtosis, mean additional delay and root mean square of delayed propagation were extracted, and maximum likelihood ratio of features was used to identify LOS and NLOS conditions [12]. However, this method is more suitable for the identification of multipath effect under NLOS, but not fully applicable to an indoor complex environment. The NLOS condition was identified by extracting kurtosis from CIR [13] and by obtaining four different features (skewness, kurtosis, RMS delay and mean excess delay) from CIR [14]. The energy of the first path in CIR was obtained as the recognition condition of NLOS [15]. Eight features were acquired from CIR [16] (standard deviation and skewness were added on the basis of literature [10]), and the convolution method was used to determine that 30 points in CIR could achieve good detection results. However, the above results did not consider the redundancy between the features, and few features would have a limitation on the detection accuracy. In addition, there is also the use of an inertial measurement unit (IMU) combined with UWB for NLOS identification [17,18], but it would utilize a second sensor.

Among the 8 CIR signal features selected in the existing research, there are often correlations between different features, and there are redundant features. Therefore, feature selection is required when identifying the propagation status of LOS and NLOS in the UWB indoor environment. Feature selection algorithms are mainly divided into two categories: one is a classifier-dependent mode (Wrapper), and the other is a classifier-independent mode (Filter) [19,20,21,22,23,24,25,26]. Compared with Wrapper mode with poor real-time performance, Filter mode has received extensive attention because it does not rely on classifiers, and it has high efficiency and scalability in reducing feature dimensions. The mutual information method in Filter mode mainly relies on the size of the mutual information value between the features and the target object to sort the features, while the classic information gain (IG) algorithm is to sort the features according to the correlation between the features and the tags [19]. A feature selection method of dynamic mutual information was proposed, which combined more information measures to construct a general criterion function [20]. However, as the feature dimension increases, the computational complexity also increases. A feature selection algorithm combining information gain and divergence was proposed for classification, but the selection index of divergence was increased, which increased the computational complexity [21]. The mutual information feature selection algorithm with uniform information distribution (MIFS-U) was proposed, which improved the penalty factor of the evaluation function in the MIFS method and used the uncertainty coefficient to describe the degree of redundancy between features, and the degree of relevance of the selected feature set and category was introduced into the penalty factor [22]. A minimum-redundancy maximum-relevance feature search algorithm and criterion (mRMR) were proposed [23]. A non-linear feature selection algorithm based on forward search was proposed [24], which used mutual information and interactive information. However, neither MIFS-U nor mRMR considers whether the first feature selected is the feature that contributes the most to the whole. An algorithm combing the mutual information criterion with the greedy search strategy was proposed, which used the nearest neighbor estimation method to estimate the mutual information and select the feature subset [25]. This algorithm has a good effect on the selection of high-dimensional features. In addition, some scholars also studied other methods for feature selection [26]. For example, a genetic algorithm was used for the CIR features of UWB selection [27]. A classical sorting algorithm ReliefF based on feature distance was introduced [28]. A backward recursive feature selection algorithm SVM recursive feature elimination (SVM-RFE) was also proposed [29]. However, the computational time complexity of these algorithms is not as good as that of feature selection methods based on mutual information. Therefore, the performance of feature selection methods based on mutual information is better for higher dimensions.

All the above methods only study the recognition of the propagation of LOS and NLOS in the indoor environment, and the influence of NLOS state on the positioning accuracy of UWB. However, the contribution of blocking category information to the perception of indoor space information is ignored, and the recognition of different blocking categories is not studied in depth. In addition, in the aforementioned CIR signal feature selection method applied to UWB, the calculation cost of different features and the number of CIR signal features are often not comprehensively considered. Some algorithms cannot achieve good feature selection results when the feature dimension is low. Meanwhile, the first feature selected by these algorithms is not the feature that contributes the most to the whole, and the computational cost of some features is often higher than other features.

### 1.2. Contributions

In order to address these problems of the above methods, this paper not only studies the recognition of LOS and NLOS states, but also considers the contribution of the effective recognition of each blocking category to indoor spatial information perception, and realizes the recognition of different blocking categories, which can be used for the indoor positioning and indoor information perception by providing more useful information. In the experiment, the blocking categories of NLOS state are expanded, including water, human body, metal, wall and wooden board, to reflect the complex indoor environment. In addition, 30 effective signal points are extracted from the CIR to improve the calculation efficiency of eight features, and the influence of different features on various shielding is analyzed.

In addition, considering the detection speed in practical application, the calculation cost of different features is introduced into the feature evaluation index. A bidirectional search algorithm based on the feature evaluation index of maximum relevance, minimum redundancy and minimum calculation cost is proposed to determine the optimal feature subset. In addition, based on the method of vector projection, the separability between the classes of measurement is realized, the decision tree support vector machine DT-SVM hierarchy is designed to complete the identification accuracy of each blocking category verification. More importantly, compared with the research of other scholars, this paper not only studies the LOS and NLOS state recognition, but also studies the recognition of different blocking categories in the NLOS state. Experiments have verified that the method proposed in this paper is superior to other algorithms in the recognition of LOS and NLOS states, but has achieved good recognition results for all blocking categories, with an average accuracy of 96.7%. This paper provides a new research method for UWB subsequent indoor positioning, indoor information perception construction and target tracking.

## 2. Experimental Environment and Channel Impulse Response (CIR) Signal Characteristic Model

### 2.1. Hardware Equipment and Experimental Environment

This test uses two UWB devices, one is a fixed base station BS as a receiving device and the other is a mobile base station MS as a transmitting device. DW1000 chip is used for each UWB base station, and the time difference of arrival (TDOA) method is used to calculate the distance. The operating frequency range of DW1000 used by BS and MS is 3244 MHz to 6999 MHz. The maximum transmission power density is −35 dBm/MHz. The communication rate is 6.8 Mbps. This experiment was conducted in a large experimental center with a relatively complicated indoor environment (listed in Figure 1). The whole room is divided into data acquisition areas for training sets and data acquisition areas for testing. In the complex indoor environment, the blocking categories usually include metal blocking (such as iron cabinets), wooden blocking (such as tables and chairs), human blocking, and water blocking (such as fish tanks and buckets of water). Therefore, in the experimental environment, the aforementioned several blocking materials were simulated to conduct experiments to verify the blocking effects of different materials on UWB. Data collection can be divided into six categories: LOS condition (no blocking) and water blocking, people blocking, metal blocking, wall blocking and wood board blocking. The physical conditions for each blocking category are as follows: In the experiment, a metal iron cabinet with 70 cm × 50 cm × 200 cm is placed between the MS and the BS, and the blocking angle of the iron cabinet is adjusted to complete the frontal blocking and the side blocking to achieve metal blocking; 1 to 2 people stand between the MS and BS at different angles to achieve human blocking. A plastic transparent water tank 80 cm × 40 cm × 80 cm, is filled with water and placed between the MS and BS to achieve water blocking. The BS is placed in the laboratory room, and the MS is placed in the corridor, separated by a concrete wall to block the wall. A wooden board 150 cm × 15 cm × 150 cm is placed between the MS and BS to block the board.

With the same distance between MS and BS, in the data collection area of the training set, each blocking category was repeatedly tested 10,000 times, and a total of 60,000 CIR samples were collected as the training set. In the data collection area of the test set, each blocking category was repeatedly tested 1000 times and a total of 6000 CIR samples were collected as the test set. The true distance between BS and MS is measured by Leica DISTO D5 laser rangefinder.

### 2.2. CIR Description

Channel state information (CSI) includes the whole process of the signal from the transmitter to the receiver, including signal fading, multipath channel, transmission path and a series of information. It is usually used to describe the transmission process of a wireless channel. CSI can be measured by channel impulse response (CIR), which is expressed as:(1)$$h(t)=\sum_{i=1}^{N}a_{i}\delta (t-\tau_{i})$$
where *N* is total of propagation paths, $a_{i}$ and $\tau_{i}$ are the amplitudes and time delays of path i, and δ is the impulse function.

The CIR signals received by BS are shown in Figure 2. It can be seen that most of the 1015 signal points are noise (blue point area), while only a few are useful information (red point area). To improve the system efficiency, only 100 points of CIR data in the red area are collected.

The 100 CIR signal points of six class data (no blocking, water blocking, people blocking, metal blocking, concrete wall blocking and wood board blocking) are shown in Figure 3. However, it can be clearly seen from Figure 3a that the CIR signal differences of several types of features only concentrate on 30 points, which is confirmed in literature [15], while the remaining points cannot contribute much difference information. Therefore, this paper extracts 30 useful signal points from the 100 points, as shown in Figure 3b.

### 2.3. Construct CIR Signal Features

By analyzing the CIR signals in the LOS and NLOS states under different blocking materials, the maximum amplitude of the signal in the NLOS state is usually smaller than that in the LOS state due to the blocking and attenuation of the signal during propagation. The energy of the first path is much less than the LOS state because of signal attenuation. In addition, the reflected, diffracted, or dispersed pulses after blocking are merged after the first path, resulting in the path amplitude after the first path in the condition of NLOS usually being larger than that in the condition of LOS, making the data below the state of NLOS closer to the mean and smaller standard deviation. At the same time, because blocking will cause delayed propagation in the NLOS state, the rise time, mean excess delay and RMS delay spread are selected as characteristics. Kurtosis and skewness are selected as characteristics due to the energy dispersion in the LOS and NLOS states. The discrete mathematical models of the above eight features are as follows:(1)The energy of the CIR signal:
(2)$$E_{f}=\sum_{t=1}^{N_{f}}h{\left(t\right)}^{2}$$
(3)$$E=\sum_{t=1}^{N}h{\left(t\right)}^{2}$$
where E and $E_{f}$ represent the energy of all CIR signals and the energy of the CIR signal reaching the first peak respectively, and N and $N_{f}$ represent all the points and the number of points reaching the first peak respectively.(2)Maximum amplitude:
(4)$$h_{\max}=\max(h(t))$$(3)The rise time to the maximum $t_{rise}$(4)The standard deviation σ(5)Mean excess delay: $\tau_{m}$
(5)$$\tau_{m}=\frac{\sum_{t=1}^{N}t|h(t){|}^{2}}{E}$$(6)RMS delay spread: $\tau_{RMS}$
(6)$$\tau_{RMS}=\sqrt{\frac{\sum_{t=1}^{N}{\left(t-\tau_{m}\right)}^{2}{\left|h(t)\right|}^{2}}{E}}$$(7)Kurtosis:
(7)$$\kappa =\frac{E[{\left(\left|h(t)\right|-u\right)}^{4}]}{\sigma^{4}}$$
where u is the mean of h(t).(8)Skewness:
(8)$$\gamma =\frac{E[{\left(\left|h(t)\right|-u\right)}^{3}]}{\sigma^{3}}$$

This paper calculates the cumulative distribution of different occlusion categories under each feature. Figure 4 shows the cumulative distribution of each feature of different occlusion categories under a 30-point CIR signal, indicating that the first path energy and the maximum amplitude under LOS status are bigger than under the NLOS condition. Rise time, standard deviation, average excess delay and delay propagation RMS under LOS status are significantly less than the NLOS condition. However, it can be seen from Figure 4 that there is no significant difference between the features of LOS and wood blocking. Meanwhile, the difference between the measured distance and the actual distance when the 15 cm thick wood is blocked is similar to the difference under LOS, which verifies that wood has little effect on the attenuation of the incident electromagnetic field when the indoor wood moisture content is low. Therefore, the following work mainly focuses on the feature selection and the best feature set of no blocking, water blocking, people blocking, metal blocking and wall blocking.

## 3. Feature Combination Selection Algorithm and Multi-Classifier Design

### 3.1. System Model Description

Given a dataset D containing k samples, $D=\{d_{1},d_{2},\ldots ,d_{k}\}$, where k represents the number of samples. In dataset D, each kind of sample $d_{i}=\{X_{i},L_{i}\}$ has feature set $X_{i}=\{x_{i1},x_{i2},\ldots ,x_{in}\}$, where n=1,2,…,8 represents the eight CIR signal: features of the first path energy, maximum amplitude, rise time, standard deviation, mean excess delay, RMS delay spread, kurtosis and skewness. $L_{i}=\{l_{i}{}^{1},l_{i}{}^{2},\ldots ,l_{i}{}^{m}\}$, where m=1,2,…,5 respectively represents no blocking, water blocking, people blocking, metal blocking and wall blocking. $l_{i}{}^{m}=\{1,-1\}$, if $l_{i}{}^{m}=1$, it is labeled as a sample of class m; otherwise $l_{i}{}^{m}=-1$. The meanings of all symbols in the article are shown in Table A1.

Define $B=\{b_{1},b_{2},\ldots ,b_{n}\}$, 1≤n≤8, *n* is the total number of selected feature; $b_{i}$, i=1,2,…,n, is the index of the feature that meet the set criteria for evaluating features.

### 3.2. Mutual Information and Relevance Definitions

The relationship between features can be divided into relevance, redundancy, interaction and independence according to mutual information. The definition of a feature’s relevance, redundancy, independence, irrelevance and interactivity is well defined in literature [25].

Mutual Information(MI) I(X;Y) is used to describe the degree of relevance between two random variables $X=\{x_{1},x_{2},\ldots ,x_{n}\}$ and $Y=\{y_{1},y_{2},\ldots ,y_{m}\}$ [19]. When I(X;Y) is large, it means the relevance between X and Y is strong; otherwise, it means the relevance is small. In particular, when I(X;Y)=0, it means that X and Y are independent of each other.
(9)$$I(X;Y)=\sum_{i=1}^{n}\sum_{j=1}^{m}p(x_{i},y_{j})\log\frac{p(x_{i},y_{j})}{p(x_{i})p(y_{i})}$$
where p(x,y) is the probability density of x and y.

Conditional mutual information (CMI) can reflect the relevance between different features under the same label. Suppose there are three random variables set X,Y and $L=\{l_{1},l_{2},\ldots ,l_{d}\}$, and their conditional probability density is p(x|l),p(y|l),p(x,y|l), respectively. In the case of a given L, then the mutual conditional information of X and Y about L is:(10)$$I(X;Y|L)=\sum_{k=1}^{d}p(l_{k})\sum_{i=1}^{n}\sum_{j=1}^{m}p(x_{i},y_{j}|l_{k})\log\frac{p(x_{i},y_{j}|l_{k})}{p(x_{i}|l_{k})p(y_{j}|l_{k})}$$

Based on the relevance between features and categories, a measurable multivariate model F can be obtained, and the specific form of F is as follows:(11)$$F=\left[\begin{array}{cccc} I_{11} & I_{12} & \cdots & I_{1m}\\ I_{21} & I_{22} & \cdots & I_{2m}\\ \vdots & \vdots & & \vdots \\ I_{n1} & I_{n2} & \cdots & I_{nm} \end{array}\right]$$
where *I* represents the mutual information between each feature and the corresponding category, and the relevance between the 8 features and the label is shown in Figure 5.

In general, the matrix $\bar{F}=[{\bar{I}}_{ij}]$ can be obtained by standardizing and centralizing the data. Calculate the feature’s mutual information matrix $Q=\bar{F}\cdot {\bar{F}}^{T}$. A n×n dimensional symmetric matrix is obtained, in which each element represents the mutual information between features under the same category, as shown in Figure 6.
(12)$${\hat{\bar{I}}}_{ij}=I_{ij}-\frac{\max I_{ij}-\min I_{ij}}{2}$$
(13)$${\bar{I}}_{ij}=\frac{{\hat{I}}_{ij}}{\sqrt{\sum_{j=1}^{m}\left({\hat{I}}_{ij}\right)}}$$
where $I_{ij}$ represents the mutual information between each feature and the corresponding category, and the relevance between the 8 features and the label is shown in Figure 6.

### 3.3. Construction of Bidirectional Search Algorithm Based on Maximum Correlation, Minimum Redundancy, and Minimum Computational Cost (mRMRMC) Feature Evaluation Criteria

The mRMR (max-relevance and min-redundancy) algorithm [23] is proposed on the basis of the mutual information-based feature selection algorithm (MIFS). The mRMR algorithm not only considers the relevance $D(x_{i},L)$ between features and labels, but also considers the redundancy $R(x_{i},x_{j})$ between features under the same label.
(14)$$D(x_{i},L)=\frac{1}{\left|L\right|}\sum_{l_{j}\in L}I(x_{i};l_{j})$$
(15)$$R(x_{i};x_{j})=\frac{1}{\left|X\right|}\sum_{x_{j}\in X}I(x_{i};x_{j})$$
where |L| represents the number of labels in tag set L, and |X| represents the number of features in feature set X.

The mRMR evaluation index is:
(16)$$J_{mRMR}(x)=\frac{1}{\left|L\right|}\sum_{l_{i}\in L}I(l_{i};x)-\frac{1}{\left|B\right|}\sum_{x_{i}\in B}I(x_{i};x)$$
where I(l;x) is the mutual information between feature x and category label l, $I(x_{i};x)$ are the mutual information between feature x and the selected feature $x_{i}$, and B represents the selected feature subset. With the increase of the number of features in the selected subsets, the calculation of the system will increase accordingly, the time cost of feature computation is introduced. Big O notation is usually used to express time complexity. Since only the highest term is retained and the coefficients are ignored, the time complexity can only be roughly described. Therefore, the calculation time complexity of each feature is expressed by calculating the number of statement execution times of each feature in this paper. Through the calculation formula of different features, the number of execution statements for each feature is obtained, and the calculation cost of each feature is obtained after ignoring the constant term.
(17)$$C(B)=\sum_{x_{i}\in B}\beta_{i}c(x_{i})$$
(18)$$\beta_{i}=\frac{c(x_{i})}{\sum_{x_{i}\in X}c(x_{i})}e^{-(\frac{D(x_{i},L)}{\sum_{x_{i}\in X}D(x_{i},L)})}$$
where C(B) represents time cost of the selected feature set B, $c(x_{i})$ represents the calculated cost of feature $x_{i}$, and $\beta_{i}$ is the relative weight of the calculation cost.

Finally, the comprehensive evaluation criteria based on maximum relevance, minimum redundancy and minimum computational cost are determined as mRMRMC:(19)$$J_{mRMRMC}(x_{i})=(\frac{1}{\left|L\right|}\sum_{l_{j}\in L}I(x_{i};l_{j})-\frac{1}{\left|B\right|}\sum_{x_{j}\in B,j\neq i}I(x_{i};x_{j})-\omega \beta_{i}c(x_{i}))$$
where $x_{i}\in B_{s}$ represents the features in the selection area; ω is an adjustable constant, indicating the weight of calculation cost in the evaluation index, and under this evaluation index, the selected set B shall be satisfied.
(20)$$B=B\cup \arg\max_{x_{i}\in B_{s}}(J_{mRMRMC}(x_{i}))$$

### 3.4. The Optimal Feature Set Is Determined Based on Multiple Threshold Constraints

In the traditional feature selection algorithm based on mutual information, such as mRMR, MIFS, IG, etc., the feature with the greatest relevance to the category is put into the selected feature subset as the first feature. However, these methods often ignore the combination effect between features, and do not consider whether the selected first feature is the feature that contributes the most to the whole. In this paper, a bidirectional search strategy based on mRMRMC, namely BS-mRMRMC algorithm, is proposed. First, calculate the relevance $D_{all}(X,L)$ of all features and labels, and then inversely delete feature $x_{i}$ to obtain the relevance $D_{\mathrm{miss}(x_{i})}(x_{i},L)$ of the feature set and labels after missing $x_{i}$. The feature that causes the greatest change in relevance is selected as the first feature vector of the selected combination. After that, the features in $B_{s}$ to be selected are selected in a positive sequence, and the features with the largest evaluation index $J_{mRMRMC}(x_{i})$ are added to the selected B, and the best feature subset is finally determined.
(21)$$D_{all}(X,L)=\frac{1}{\left|L\right|}\sum_{x_{i}\in X}\sum_{l_{j}\in L}I(x_{i};l_{j})$$
(22)$$D_{miss(x_{i})}=\frac{1}{\left|L\right|-1}\sum_{x_{j}\in X,x_{j}\neq x_{i}}\sum_{l_{k}\in L}I(x_{j};l_{k})$$

If $x_{i}=\arg\max(D_{all}(X,L)-D_{miss(x_{i})}(x_{i},L))$, then feature $x_{i}$ provides the most information for the whole, so $x_{i}$ is taken as the first feature of the selected set, $B(1)=x_{i}$.

Reverse deletion ensures that the first feature selected must be the feature that provides the most information for the whole, avoids that the features which are most relevant to the tag are not the feature that provides the most information for the whole due to the combination effect.

This paper makes the system model more efficient and stable by adding three constraint conditions: minimum relevance constraint, computational cost constraint and maximum evaluation index constraint. The selection of the optimal feature subset based on BS-mRMRMC algorithm under three constraints is shown in Figure 7.

Constraint 1: minimum relevance constraint. Set the threshold of relevance between feature and label $\eta_{d}$. Once the feature is related to the label $D(x_{i},L)<\eta_{d}$, the feature $x_{i}$ is directly discarded to reduce the complexity of later calculation and improve the system efficiency. Where $\eta_{d}$ is selected according to the relevance between the global feature vector and the label:(23)$$\eta_{d}=\lambda_{d}\max\{D(x_{i};L),x_{i}\in X\}$$
where $0<\lambda_{d}\leq 1$ represents the parameter of minimum relevance threshold.

Constraint 2: computational cost constraint. Set the computational cost constraint threshold $\eta_{c}$, $x_{j}=\arg\max_{x_{i}\in B_{s}}(J_{mRMRMC}(x_{i}))$, if $C(B)+c(x_{j})>\eta_{c}$, then the best subset of features is directly determined to be $B_{best}=B$. The computing cost constraint threshold $\eta_{c}$ is determined according to the complexity of computing system characteristics and the computing power of the system in practical application.
(24)$$\eta_{c}=\lambda_{c}\sum_{x_{i}\in X}c(x_{i})$$
where $0<\lambda_{c}\leq 1$ represents the parameter of computing cost threshold.

Constraint condition 3: maximum evaluation index constraint. Set the maximum evaluation index constraint threshold $\eta_{j}$, Indicates that the recognition accuracy of the system can reach the actual requirement $\eta_{j}$ after $x_{j}$ is added, the best subset of features $B_{best}=B\cup x_{j}$.

### 3.5. Decision Tree Support Vector Machine (DT-SVM) Classifier Design

SVM, one of the most commonly used classifiers, was proposed by Vapnik et al. based on statistical learning theory and its learning method [30]. Due to its strong learning ability and generalization ability, SVM has been widely used in multi-classification problems with small samples. However, SVM was originally designed to handle two types of classification tasks. For multi-classification support vector machines, N-class tasks can be converted into multiple two-class tasks as in one-against-all SVM (OAA-SVM) [31], one-against-one SVM (OAO-SVM) [32], decision directed acyclic graph SVM (DDAG-SVM) [33], and the multi-classification method decision tree SVM (DT-SVM) [34]. For N-class classification problems, DDAG-SVM and DT-SVM only need to construct the decision surface, which greatly improves the training and detection speed. But the recognition accuracy of DDAG-SVM and DT-SVM is not ideal due to the accumulation of classification errors. In order to solve this problem, both [35] and [36] proposed the use of the particle swarm optimization algorithm DT-SVM. Although this method improves the classification accuracy, it also increases the corresponding computational complexity and has a good effect on more classification. The structure of DT-SVM classifier is designed by different methods [37,38,39,40], but there are always problems such as limited accuracy improvement and limited adaptability.

Therefore, considering the actual number of categories in this paper, the vector projection method to realize the separability measurement between classes is adopted. Accordingly, the hierarchical structure of the DT-SVM classifier is designed to realize the recognition of each blocking category. The method based on vector projection was proposed by Li et al. [41], by which the number of intersecting samples between two types of samples could be better distinguished. The method based on vector projection was proposed by Li et al. [41], by which the number of intersecting samples between two types of samples could be better distinguished.

Set the sample set $D_{l}=\{d_{l}{}_{1},d_{l2},\ldots ,d_{li}\}$ of class l, indicating that there are $k_{l}$ samples of class l. $d_{li}=\{x_{i1},x_{i2},\ldots ,x_{ij}\}$, j=1,2,…,n, indicating that each sample contains n features. Let: (25)$$m_{l}=\frac{1}{k_{l}}\sum_{i=1}^{k_{l}}d_{li}$$
where ${d^{\prime }}_{li}$ is the projection of sample $d_{li}$ to the feature direction of this class, Euclidean distance from ${d^{\prime }}_{li}$ to $m_{l}$ is ${\left\|m_{l}{d^{\prime }}_{li}\right\|}_{2}={\left\|m_{l}-{d^{\prime }}_{li}\right\|}_{2}$.

Let:$d={\left\|m_{1}-m_{2}\right\|}_{2}$ represents the distance between the two classes $D_{1}=\{d_{1}{}_{1},d_{12},\ldots ,d_{1i}\}$
$D_{2}=\{d_{2}{}_{1},d_{22},\ldots ,d_{2i}\}$. And
(26)$$\{\begin{array}{c} r_{1}=\underset{d_{1i}\in D_{1}}{\max}({\left\|m_{1}{d^{\prime }}_{1i}\right\|}_{2})\\ r_{2}=\underset{d_{2i}\in D_{2}}{\max}({\left\|m_{2}{d^{\prime }}_{2i}\right\|}_{2}) \end{array}$$

In order to measure the separability between classes, the separability measure value $se_{12}$ is calculated. If $r_{1}+r_{2}\geq d$, the number of samples in $X_{1}$ that satisfy $d-r_{2}\leq {\left\|m_{1}{d^{\prime }}_{1i}\right\|}_{2}\leq r_{1}$ and ${\left\|m_{1}{d^{\prime }}_{1i}\right\|}_{2}\leq d$ is $n_{1}$. In the same way, the number of samples in $X_{2}$ that satisfy $d-r_{1}\leq {\left\|m_{2}{d^{\prime }}_{2i}\right\|}_{2}\leq r_{2}$ and ${\left\|m_{2}{d^{\prime }}_{2i}\right\|}_{2}\leq d$ is $n_{2}$.Then, the separability measure between classes is defined as $se_{12}$.
(27)$$se_{12}=\frac{n_{1}+n_{2}}{k_{1}+k_{2}}$$

If $r_{1}+r_{2}<d$, that means the two classes don’t intersect, $se_{12}=0$.


**Step1:** Let d=1, and the separable measure $se_{ij}$ between classes was obtained from the class samples, i,j=1,2,…,l,i≠j, and the separable measure matrix SE was constructed.
(28)$$SE=\left[\begin{array}{ccccc} 0 & se_{12} & \cdots & se_{1,l-1} & se_{1,l}\\ se_{21} & 0 & \cdots & se_{2}{}_{,l-1} & se_{2,l}\\ \vdots & \vdots & \ddots & \vdots & \vdots \\ se_{l-1,1} & se_{l-1,2} & \cdots & 0 & se_{l-1,c}\\ se_{l,1} & se_{l,2} & \cdots & se_{l,l-1} & 0 \end{array}\right]$$**Setp2**: according to the sum of each row, find the trip and the smallest row, and record the number of rows at this time L(d)=m+d−1, then delete the elements of that row and column.**Step3**: d=d+1, and repeat step2 to guide the sorting of inter-class relationships of all categories.**Step4**: Initialize d=1, and take the sample set of class d=1 as the positive sample of subclassifier, and the remaining sample of classes as the negative sample of the classifier. Train the SVM classifier and record the node information node(d). Remove the class L(d) sample set from the sample set.**Step5**: d=d+1, and repeat step4 until all sub-classifiers are trained. The final DT-SVM multi-classifier is shown in Figure 8.


The flow chart of the whole paper is addressed in Figure 9. Firstly, data collection and data preprocessing were carried out, and then the best feature subset was obtained based on the BS-mRMRMC algorithm. Finally, the classification hierarchy of multiple classifiers was determined according to the vector projection method, and the performance of the classifier was evaluated with four indexes.

## 4. Results and Discussion

In this section, the effectiveness of the proposed method is verified by the selection effect of features in the best subset and the evaluation indexes of recognition results. As shown in Figure 10, three feature selection algorithms IG [20], ReliefF [27] and SVM-RFE [29] are selected to compare the results with the BS-mRMRMC algorithm proposed in this paper.

In this paper, four system performance evaluation indexes are selected to evaluate the results of each index under different feature subsets selected by four methods. The meanings of several parameters of performance evaluation are shown in Table 1.

Four evaluation indexes are overall accuracy, precision, recall rate and the comprehensive evaluation index. The overall accuracy can evaluate the positive and negative samples detection accuracy, precision can evaluate the misjudgment of positive samples of the system, and the recall rate can evaluate the missed sanction of positive samples of the system, and the comprehensive evaluation index can comprehensively evaluate the identification performance of the system.
(1)Overall accuracy:
(29)$$OA=\frac{TP+TN}{TP+FN+TN+FP}$$(2)Precision:
(30)$$P=\frac{TP}{TP+FP}$$(3)Recall:
(31)$$R=\frac{TP}{TP+FN}$$(4)Comprehensive evaluation index F-Score:
(32)$$F=\frac{(\alpha^{2}+1)P*R}{\alpha^{2}(P+R)}$$
Usually take α=1, $F1=\frac{2*P*R}{P+R}$.

The feature ordering of the four methods adopted in this paper is shown in Table 2. It can be analyzed from the feature selection in the table that the feature with the greatest relevance with the label is not necessarily the feature that contributes the most to the overall information, which also verifies the effectiveness of the paper’s reverse deletion search strategy used to determine the first feature.

### 4.1. Best Subset Selection Based on Different Constraints

First, the paper verifies the number of features in the optimal feature subset determined by each method under different constraints of computational cost and maximum detection accuracy. Different $\eta_{c}$ and $\eta_{j}$ are obtained by setting different threshold parameters of calculation cost and maximum evaluation index $\lambda_{c}=[0.1,0.2,\ldots ,1]$, $\lambda_{j}=[0.5,0.55,\ldots ,0.95]$. From Figure 11, the number of features in the best feature subset selected by the BS-mRMRMC method is less than that of other methods under higher detection accuracy requirements and lower computational cost requirements. That is to say, the method in this paper can achieve the same or higher detection accuracy with fewer features. For example, when the constraints are $\eta_{c}\leq 0.5C(X)$ and $\eta_{j}\geq 0.9$, the number of best sub-sets selected by the proposed algorithm is only 2, when the constraints are changed by $\eta_{c}\leq 0.5C(X)$ and $\eta_{j}\geq 0.95$, the number is only 3. However, the number of features selected by other algorithms under the same constraint conditions are more than the algorithm of this paper. It can be concluded that the number of features determined by the best subset of the proposed algorithm is better than that of the other three algorithms when the computational cost is lower and the demand for higher precision is higher.

### 4.2. The Results Were Compared and Analyzed with Different Feature Number

In this section, the paper studies the comparison of the recognition results of different categories at first according to the sorting results of features in Table 2. The four pictures of Figure 12 show the comparison surface diagram of four evaluation indexes for identifying each category by four methods, when one to eight different features are selected successively. Figure 12a shows that our algorithm can achieve higher detection accuracy with fewer features. When the number of features selected is less than 4, the identification accuracy of each category is significantly better than IG and SVM-RFE, and slightly better than ReliefF. The Figure 12b–d show that our method has better recognition accuracy, recall rate and comprehensive evaluation index for most categories than the other three algorithms when the number of selected features is less than 4. This result is also consistent with that in the previous section, under the constraint of $\eta_{c}\leq 0.5C(X)$ and $\eta_{j}\geq 0.95$, the algorithm in this paper selects only 3 optimal subsets, which can achieve higher detection accuracy at a lower computational cost with a smaller number of features.

Figure 13 shows average accuracy rate, average precision, average recall rate and average comprehensive evaluation index contrast line chart of four methods when selecting a different number of features. If the number of features selected is less than 4, the indicators of the BS-mRMRMC algorithm are better than other algorithms. When there are more features, the selection of feature sets of different algorithms for the final indicators tend to be consistent. Due to the redundancy between features not considered by the IG algorithm, the detection accuracy decreases when the third feature is added in Figure 13, which also verifies the importance of considering the redundancy between features. In addition, the first feature selected can be regarded as the feature with the greatest contribution to overall performance. The accuracy of the recognition under the first feature is better than other algorithms, which also verifies the effectiveness of the bidirectional search strategy. However, the average recognition accuracy of the first feature selected by the BS-mRMRMC method is high, but the average recall rate is poor, indicating that this feature has a great correlation with other categories, which can be verified from Figure 5. As a result, many positive samples are missed and identified as negative samples, and there is a certain contradiction between the recall rate and accuracy. Therefore, it can be seen from the average comprehensive evaluation index F1 obtained from the comprehensive accuracy and recall rate that the first feature selected by BS-mRMRMC is better.

The purpose of feature selection is to obtain the best and smallest feature set, and the system performance meets the constraint requirements. Compared with the ReliefF method, our algorithm can comprehensively consider the correlation between features and categories and the redundancy between features. The limitation of ReliefF is that it cannot effectively remove redundant features. It can also be seen from Figure 13 that our proposed method selects three features, but ReliefF needs to select five features to achieve the same index requirements. Under the same constraints, our algorithm can select the feature set with the largest correlation, the smallest redundancy, and a smaller number of features. When the number of features in the selected best feature subset is 3, comparison of identification indicators of each category under the four methods is shown in Table 3. Except for the fact that the recognition index of some categories is slightly lower than other algorithms, the recognition index of BS-mRMRMC algorithm of most categories is better than other algorithms, and the average index of each category is more than 95%.

## 5. Conclusions

In this paper, a large amount of CIR data of a UWB device with five different blocking materials at LOS and NLOS are obtained in a large experimental center, and eight features are extracted from CIR to analyze the influence of different features on different occlusion categories. Selecting 30 effective CIR signal points for feature extraction can effectively improve the calculation efficiency of feature extraction. The proposed BS-mRMRMC method introduces the computational cost into the mRMRMC feature evaluation criteria, and uses a bidirectional search strategy for feature selection. Through multiple experiments, it has been proved that under the same constraints, BS-mRMRMC can better select the optimal feature subset with maximum correlation, minimum redundancy, minimum computational cost and fewer features. Meanwhile, when the same number of features are selected, the BS-mRMRMC method has better recognition accuracy for each category than the other three methods. When only three features are selected, the average accuracy of the recognition for each blocking category can reach 96.7%. More importantly, this paper not only identifies the LOS and NLOS categories, but also identifies the different blocking categories under NLOS, which provides a new research method for UWB subsequent indoor positioning, indoor information perception construction and target tracking.

## Figures and Tables

**Figure 1 sensors-20-04178-f001:**
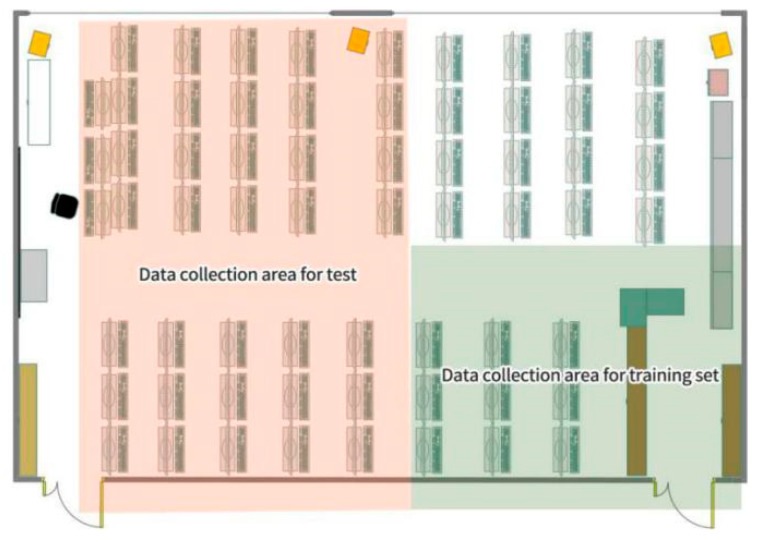
Experimental test environment.

**Figure 2 sensors-20-04178-f002:**
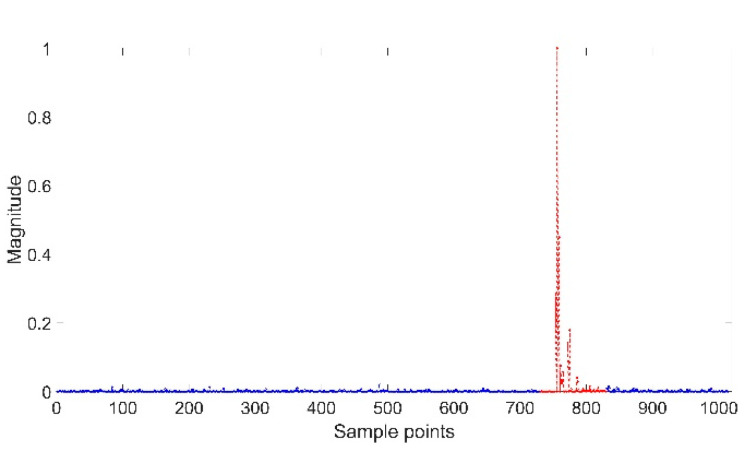
1015 CIR signal points.

**Figure 3 sensors-20-04178-f003:**
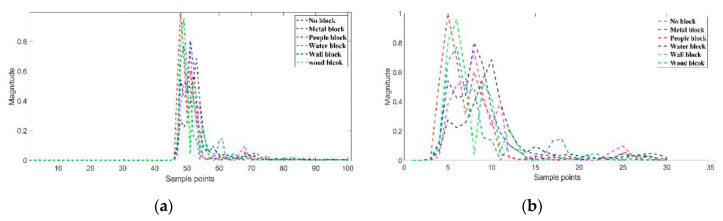
The different channel impulse response (CIR) signal points of 6 categories: (**a**) 100 CIR signal points of 6 categories; (**b**) 30 CIR signal points of 6 categories.

**Figure 4 sensors-20-04178-f004:**
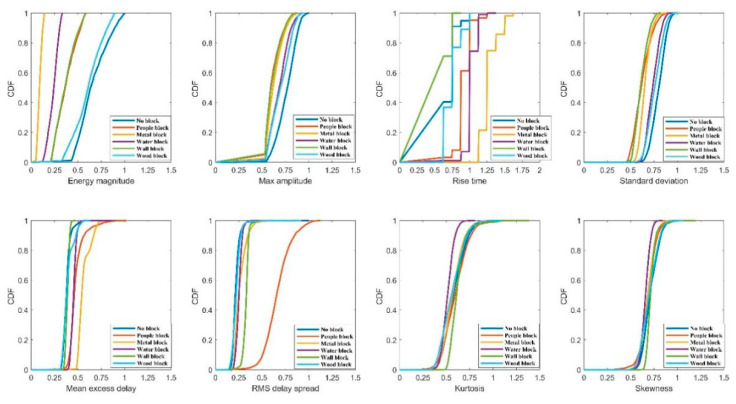
The cumulative distribution of features of different blocking classes at 30 CIR signal points.

**Figure 5 sensors-20-04178-f005:**
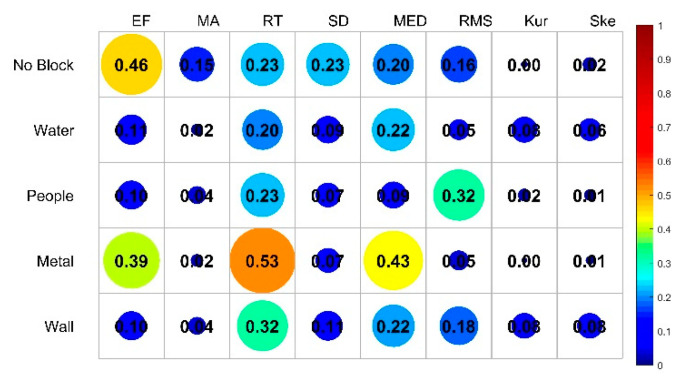
The relevance between features and labels.

**Figure 6 sensors-20-04178-f006:**
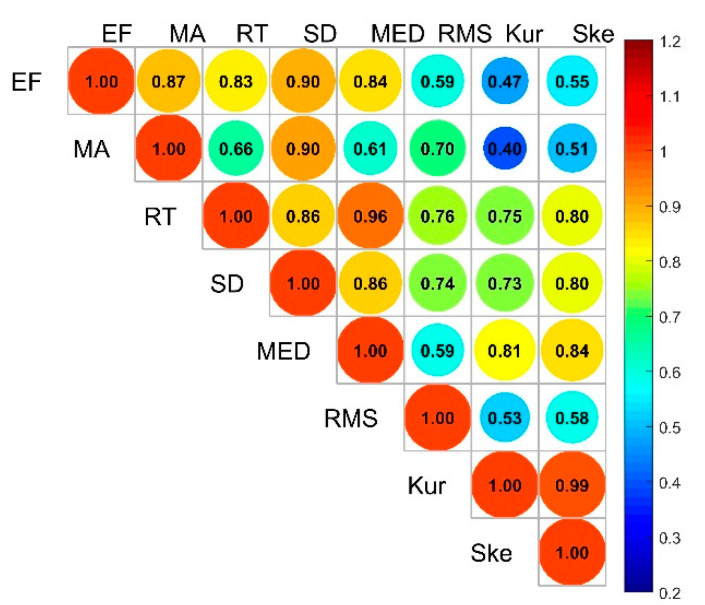
The relevance between features and features under the same label.

**Figure 7 sensors-20-04178-f007:**
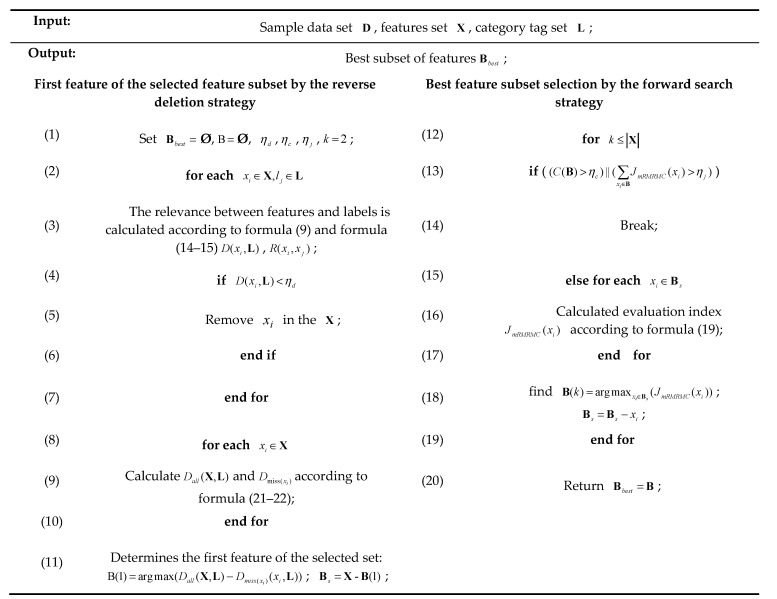
The solution process of the proposed algorithm.

**Figure 8 sensors-20-04178-f008:**
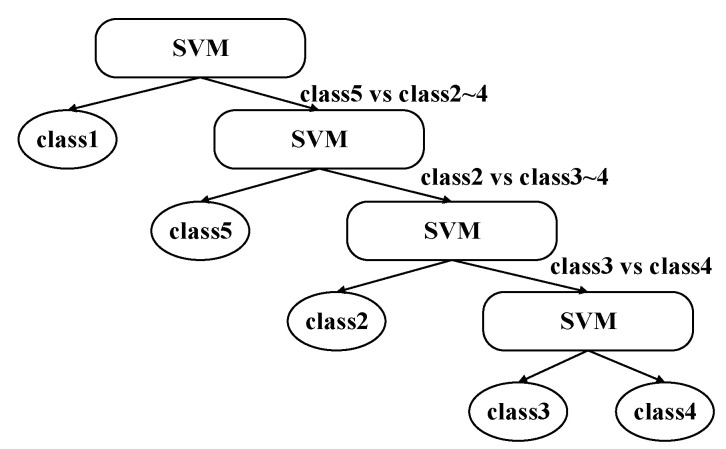
Hierarchical structure of decision tree support vector machine (DT-SVM) multi-classifier.

**Figure 9 sensors-20-04178-f009:**
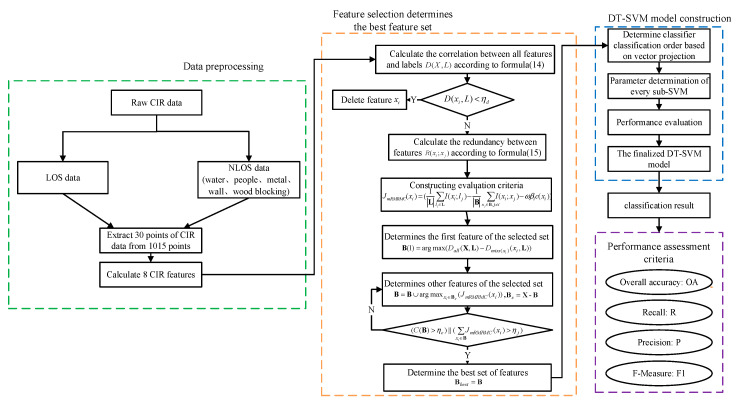
General diagram of algorithm flow.

**Figure 10 sensors-20-04178-f010:**
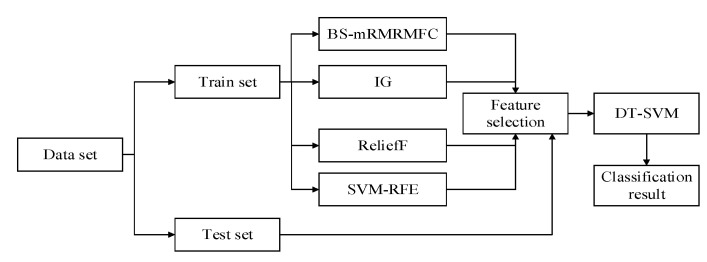
Experimental framework.

**Figure 11 sensors-20-04178-f011:**
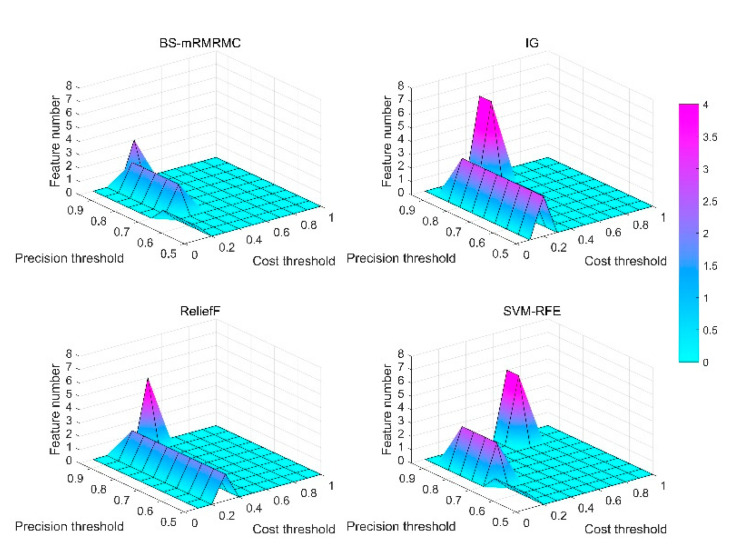
Surface graphs of the number of feature of best feature subset for the four methods under different computational cost constraints $\eta_{c}$ and maximum evaluation index constraints $\eta_{j}$.

**Figure 12 sensors-20-04178-f012:**
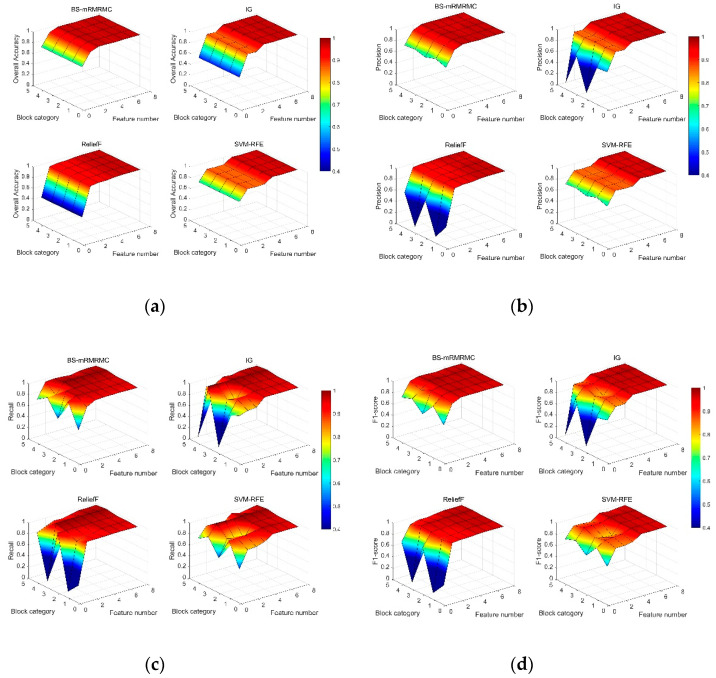
Under different number of features, the evaluate indicator surface graph of the four methods identify each category. (**a**) The overall accuracy, (**b**) the accuracy, (**c**) the recall rate, and (**d**) the comprehensive evaluation indicator f1-measure; where block category 0,…,4 respectively represents no blocking, water blocking, people blocking, metal blocking and wall blocking).

**Figure 13 sensors-20-04178-f013:**
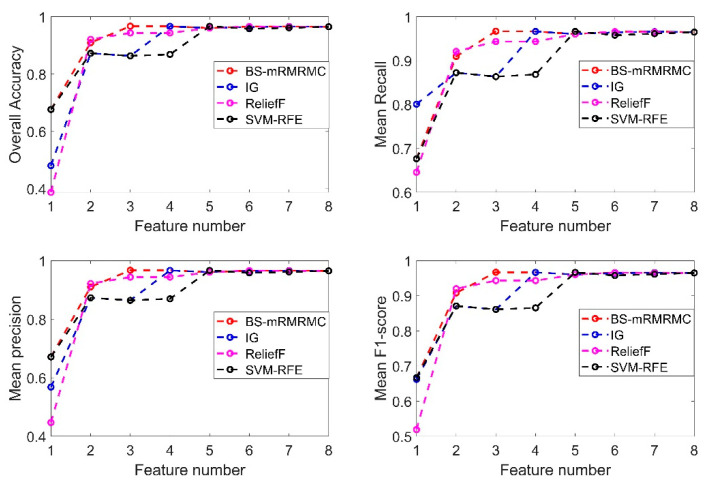
Comparison diagram of average indexes of each category identified by four methods under different feature quantities.

**Table 1 sensors-20-04178-t001:** Definition table of performance evaluation parameters.

Actual class	**Predicted Class**			
		Yes	No	Total
	Yes	TP	FN	P
	No	FP	TN	N
	Total	P’	N’	P+N=P’+N’

**Table 2 sensors-20-04178-t002:** Feature selection ranking of four algorithms.

Algorithms	Feature Selection Sorting
BS-mRMRMC	$t_{rise}>\tau_{RMS}>E_{f}>\tau_{m}>\sigma >\kappa >h_{\max}>\gamma$
IG	$E_{f}>t_{rise}>\tau_{m}>\tau_{RMS}>\sigma >h_{\max}>\gamma >\kappa$
ReliefF	$\tau_{RMS}>E_{f}>\tau_{m}>\sigma >t_{rise}>\gamma >\kappa >h_{\max}$
SVM-RFE:	$t_{rise}>E_{f}>\tau_{m}>\gamma >\kappa >\tau_{RMS}>h_{\max}>\sigma$

**Table 3 sensors-20-04178-t003:** The identification indexes of various categories are compared under the four methods.

	Precision				Recall				F1-Score				Overall Accuracy			
Method	BS-mRMRMC	IG	ReliefF	SVM-RFE	BS-mRMRMC	IG	ReliefF	SVM-RFE	BS-mRMRMC	IG	ReliefF	SVM-RFE	BS-mRMRMC	IG	ReliefF	SVM-RFE
**No block**	97.28%	86.85%	94.41%	86.85%	94.00%	80.50%	94.00%	80.50%	95.61%	83.55%	94.21%	83.55%	96.70%	86.35%	94.35%	86.35%
**Water block**	96.69%	85.90%	94.98%	85.90%	96.75%	92.50%	91.00%	92.50%	96.72%	89.08%	92.95%	89.08%	96.70%	86.35%	94.35%	86.35%
**People block**	96.59%	87.57%	93.97%	87.57%	97.25%	73.50%	96.50%	73.50%	96.92%	79.92%	95.22%	79.92%	96.70%	86.35%	94.35%	86.35%
**Metal block**	96.14%	85.45%	93.48%	85.45%	99.50%	99.50%	99.50%	99.50%	97.79%	91.94%	96.40%	91.94%	96.70%	86.35%	94.35%	86.35%
**Wall block**	96.85%	86.40%	95.03%	86.40%	96.00%	85.75%	90.75%	85.75%	96.42%	86.07%	92.84%	86.07%	96.70%	86.35%	94.35%	86.35%
**Average**	96.71%	86.43%	94.37%	86.43%	96.70%	86.35%	94.35%	86.35%	96.69%	86.11%	94.32%	86.11%	96.70%	86.35%	94.35%	86.35%

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
