# Peer review of "Recognition of Blocking Categories for UWB Positioning in Complex Indoor Environment"

_sensors, 2020, doi:10.3390/s20154178_

Round 1

Reviewer 1 Report

In this paper, the authors propose a method that detects LOS and NLOS environments as well as blocking categories for NLOS environment. The proposed method is based on BS-mRMRMC algorithm with DT-SVM classifier. The proposed method seems quiet useful, but the following points should be resolved for publication.

1. Experimental environment

- The authors should provide the following experimental conditions:

1) Basic specifications including an operating frequency and transmitting power of the BS and MS

2) Repetition number of the CIR measurement for the same blocking category

3) Physical conditions of each blocking category (e.g. how many people block the LOS path and where those people are located for the people block category; how large wood is located on the LOS path for the wood block category)

- In the experiment, the measured data is ignored when the difference between the measured and true distances does not meet a specified condition (e.g. the difference is greater than 20 cm in LOS condition). The author should justify the distance condition by explaining why the measured data should be ignored the specific difference.

- The authors should justify the utilization of the block categories. Among the block categories, the water block seems unrealistic especially. Is it realistic to encounter the water block in an indoor environment?

2. Algorithm design

- The authors should present a criterion to determine N and Nf in (2) and (3). How can be the N and Nf determined? The Nf represents the number of points reaching the first path, then is the Nf always equal to 1?

- Fig. 4 shows that there is no significant difference between the LOS and wood blocking cases. This means that the wood block is insignificant on the LOS path. The author justifies this results. Usually, a wood material contains high moisture inside, so attenuates significantly an incident electromagnetic field.

- Among the contributions of this paper, the calculation cost is introduced in the BS-mRMRMC algorithm. However, it is hard to understand how the cost is considered for the determination of the evaluation criteria. The author should present the cost function in detail after the equation (18).

3. Comparison with other algorithms

- Fig. 12 shows that the proposed BS-mRMRMC algorithm has higher accuracy when the feature number is less than 4. The BS-mRMRMC algorithm, however, has inferior performance in terms of the mean recall to other algorithms when the feature number is 1. The authors should explain why only the mean recall is not good in the BS-mRMRMC algorithm. 

- Like the IG algorithm, the SVM-RFE algorithm shows that the accuracy rather decreases when the feature number increases from 2 to 3. Is this related to the redundancy problem?

- In Fig. 12, the ReliefF algorithm has similar accuracies as the BS-mRMRMC algorithm when the feature number is over 3. This results imply that the proposed algorithm has a similar accuracy if the feature number is enough. The author should present an advantage of the proposed method especially in comparison with the ReliefF algorithm.   

Author Response

Thank you very much for your suggestions and questions.

For each of your suggestions and questions, we have answered them and added amendments to the manuscript.

  1. Experimental environment

1)The authors should provide the following experimental conditions:

Basic specifications including an operating frequency and transmitting power of the BS and MS

Answer:

The specifications of MS and BS have been supplemented in lines 138-140 of the article.

The operating frequency range of DW1000 used by BS and MS is 3244MHz to 6999MHz. The maximum transmission power density is -35dBm/MHz. The communication rate is 6.8Mbps.

2) Repetition number of the CIR measurement for the same blocking category

Answer:

This point is supplemented in lines 158-161 of the article.

With the same distance between MS and BS, in the data collection area of the training set, each blocking category was tested repeatedly 10000 times, a total of 60,000 CIR samples were collected as the training set, and in the data collection area of the test set Each blocking category in the test was repeated 1000 times and a total of 6,000 CIR samples were collected as the test set.

3)Physical conditions of each blocking category (e.g. how many people block the LOS path and where those people are located for the people block category; how large wood is located on the LOS path for the wood block category)

Answer:

This point is supplemented in lines 149-157 of the article:

Metal blocking: The physical conditions for each blocking category are as follows: In the experiment, a metal iron cabinet with 70cm*50cm*200cm(L*W*H) was placed between the MS and the BS, and the blocking angle of the iron cabinet was adjusted to complete the frontal blocking and the side blocking to achieve metal blocking.

People blocking: 1 to 2 people stand between the MS and BS at different angles to achieve human blocking.

Water blocking: A plastic transparent water tank is 80 cm*40 cm*80 cm (L*W*H) , was filled with water and placed between the MS and BS to achieve water blocking.

Wall blocking: The BS was placed in the laboratory room, and the MS was placed in the corridor, separated by a concrete wall to block the wall.

Wood blocking: A wooden board 150 cm*15 cm *150 cm (L*W*H)  was placed between the MS and BS to block the board.

4) In the experiment, the measured data is ignored when the difference between the measured and true distances does not meet a specified condition (e.g. the difference is greater than 20 cm in LOS condition). The author should justify the distance condition by explaining why the measured data should be ignored the specific difference.

Answer:

Due to the relatively complicated indoor experimental environment, the data measured through multiple experiments, under the LOS state, UWB's ranging error is usually within 10cm, but due to the long time of the measured data, there are laboratory personnel walking in the middle (such as from MS Walk with BS), resulting in a sudden increase in ranging error. When we collect data, we mark the time point of the abnormal situation, and find that the error is greater than 20cm when the data is abnormal, so we delete the data in the marked time node.

In the process of collecting NLOS state data, due to the need to adjust the number and position of obstructions in the middle (for example, when two people turn around, causing a large gap between the two people), some LOS state data will be collected. Data with a distance error less than 10cm appears within the marked time node range, so data with a distance error less than 10cm within the marked time node range in the NLOS state is filtered.

The above work is to deal with the abnormal data when collecting data to ensure the validity and accuracy of the original data, but the 20cm and 10cm here be related to the experimental environment and are not of generalized significance, so this part is unnecessary to explain in the paper .

5) The authors should justify the utilization of the block categories. Among the block categories, the water block seems unrealistic especially. Is it realistic to encounter the water block in an indoor environment?

Answer:

This point is supplemented in lines 143-147 of the article.

In the complex indoor environment, the blocking category usually includes metal blocking such as iron cabinets, wooden blocking such as tables and chairs, human blocking, and water blocking such as fish tanks and buckets of water. Therefore, in the experimental environment, the above-mentioned several blocking materials were simulated to conduct experiments to verify the blocking effects of several materials on UWB.

  1. Algorithm design

1) The authors should present a criterion to determine N and Nf in (2) and (3). How can be the N and Nf determined? The Nf represents the number of points reaching the first path, then is the Nf always equal to 1?

Answer:

This point is supplemented in lines 297-201 of the article.

The definition of N and Nf is not explained clearly here. The energy of the first path here refers to the sum of the discrete points of CIR at all times before the UWB CIR signal reaches the first peak. So N and Nf respectively represent all the points and the number of points reaching the first peak.

2) - Fig. 4 shows that there is no significant difference between the LOS and wood blocking cases. This means that the wood block is insignificant on the LOS path. The author justifies this results. Usually, a wood material contains high moisture inside, so attenuates significantly an incident electromagnetic field.

Answer:

This point is supplemented in lines 215-218 of the article.

That's right, when the moisture content of the indoor wood is high, the wood has a great influence on the attenuation of the incident electromagnetic field. However, the wooden boards under normal indoor environment selected in our experiment usually contain less moisture, and the measured data and analysis prove that the 15cm thick wooden board has a little impact on UWB's distance measurement when the internal moisture content is small.

3)- Among the contributions of this paper, the calculation cost is introduced in the BS-mRMRMC algorithm. However, it is hard to understand how the cost is considered for the determination of the evaluation criteria. The author should present the cost function in detail after the equation (18).

Answer:

This point is supplemented in lines 273-279 of the article.

Big O notation is usually used to express time complexity. Since only the highest term is retained and the coefficients are ignored, the time complexity can only be roughly described. Therefore, the calculation time complexity of each feature is expressed by calculating the number of statement execution times of each feature in this paper. Through the calculation formula of different features, the execution statement times of each feature is obtained, and the calculation cost of each feature is obtained after ignoring the constant term.

  1. Comparison with other algorithms

1) - Fig. 12 shows that the proposed BS-mRMRMC algorithm has higher accuracy when the feature number is less than 4. The BS-mRMRMC algorithm, however, has inferior performance in terms of the mean recall to other algorithms when the feature number is 1. The authors should explain why only the mean recall is not good in the BS-mRMRMC algorithm. 

Answer:

This point is supplemented in lines 453-459 of the article.

he average recognition accuracy of the first feature selected by the BS-mRMRMC method is high, but the average recall rate is poor, indicating that this feature has a great correlation with other categories, which can be verified from Figure 5. As a result, many positive samples are missed and identified as negative samples, and there is a certain contradiction between the recall rate and accuracy. Therefore, it can be seen from the average comprehensive evaluation index F1 obtained from the comprehensive accuracy and recall rate that the first feature selected by BS-mRMRMC is better.

2)- Like the IG algorithm, the SVM-RFE algorithm shows that the accuracy rather decreases when the feature number increases from 2 to 3. Is this related to the redundancy problem?

Answer:

Yes, the third feature selected by the IG and SVM-RFE methods has a great correlation with the category, but the feature has greater redundancy with the first two selected features. Because these two algorithms do not Considering redundancy between features, leading to the addition of a feature with great relevance to the category will lead to a reduction in recognition accuracy, which also reflects the importance of considering feature redundancy.

3)- In Fig. 12, the ReliefF algorithm has similar accuracies as the BS-mRMRMC algorithm when the feature number is over 3. This results imply that the proposed algorithm has a similar accuracy if the feature number is enough. The author should present an advantage of the proposed method especially in comparison with the ReliefF algorithm.

Answer:

This point is supplemented in lines 467-475 of the article.

The purpose of feature selection is to produce the best and smallest feature set, so that the system performance meets the constraint requirements. Compared with the ReliefF method, our algorithm can comprehensively consider the correlation between features and categories and the redundancy between features. However, the limitation of ReliefF is that it cannot effectively remove redundant features, so it can also be seen from Figure 12, proposed method selects three features, but ReliefF needs to select five features to achieve the same index requirements. Under the same constraints, our algorithm can select the feature set with the largest correlation, the smallest redundancy, and a smaller number of features.

Thank you again for your review, we will make timely additions and amendments if there are any problems.

Reviewer 2 Report

1)The abbreviations in the abstract should be supplemented by the corresponding full names;

2)The format of the text and paragraph in the introduction is incorrect;

3)It is unreasonable to use Arabic numerals to represent quantity;

4)Is the reference 10 in the sentence "standard deviation and skewness were added on the basis of literature 9"?

5)There are grammatical errors in the Line 66 and 77;

6)What is the didifference between MIFS-U and MIFS;

7)What does two algorithms mean?

8) Authors state that "the time difference of arrival (TDOA) method is used to calculate the distance". Is the TDOA method used for positioning?Is TOF ranging used in UWB DW1000?

9)Is the data acquisition connected to the base station via USB?

10)The real experimental scenario should be shown;

11)The novelty needs further explanation;

12)The reference format is not standard.

Author Response

Thank you very much for your suggestions and questions.

For each of your suggestions and questions, we have answered them and added amendments to the manuscript.

  • 1)The abbreviations in the abstract should be supplemented by the corresponding full names;

Answer:

Thanks for your correction, the 8 to 21 lines have been modified as required.

  • 2)The format of the text and paragraph in the introduction is incorrect;

Answer :

The format of the text and paragraph in the introduction have been modified as required

  • 3)It is unreasonable to use Arabic numerals to represent quantity;

Answer:

The 49 line and 62 line have been modified as required.

  • 4)Is the reference 10 in the sentence "standard deviation and skewness were added on the basis of literature 9"?

Answer:

The 63 line have been modified as required

Here is the writing error, thank you for your discovery, it should be reference 10 that standard deviation and skewness were added on the basis of literature 10.

  • 5)There are grammatical errors in the Line 66 and 77;

Answer:

The 64 to 66 lines have been modified as required.

  • 6)What is the didifference between MIFS-U and MIFS;

Answer:

The 83 to 86 lines have been modified as required.

The penalty factor in the evaluation function in the MIFS selection algorithm does not accurately express the increase in the degree of redundancy. The mutual information feature selection algorithm with uniform information distribution (MIFS-U) improves the penalty factor of the evaluation function in the MIFS method, and uses the uncertainty coefficient to describe the degree of redundancy between features, and the degree of relevance of the selected feature set and category is introduced into the penalty factor

  • 7)What does two algorithms mean?

Answer:

The 89 line have been modified as required.

These two algorithms are MIFS-U and mRMR, These two algorithms consider the correlation between features and categories and the redundancy of features and selected features, but do not consider whether the first feature selected is the feature that contributes the most to the whole, and the calculation cost of each feature is not considered. So we introduce the computational cost of features to construct an evaluation function and use a bidirectional search strategy to determine the first feature.

  • 8)Authors state that "the time difference of arrival (TDOA) method is used to calculate the distance". Is the TDOA method used for positioning?Is TOF ranging used in UWB DW1000?

Answer:

We use the TDOA method to measure the distance between MS and BS, and UWB positioning requires at least three base stations ranging to solve to calculate the position of MS. Because TDOA does not directly use the signal arrival time, but uses the time difference of multiple base stations to receive the signal to determine the position of the moving target, there is no need to add a special timestamp to synchronize the clock, the positioning accuracy is relatively improved, but each in the system Anchor's clock must be strictly synchronized. TOF ranging does not neet the time synchronization between the base station and the tag, but the time of the TOF ranging method depends on the clock accuracy, and clock offset will cause errors. So we use the TDOA method.

  • 9)Is the data acquisition connected to the base station via USB?

Answer:
Data collection can be done through UWB or WIFI, our data collection is through USB.

  • 10)The real experimental scenario should be shown;

Answer:

Since no pictures were taken during the experiment, and now we could not return to the school to take pictures because of the novel coronavirus, temporarily unable to provide live photos of the experimental environment, but the experimental environment and physical experimental conditions are described in detail in lines 138 to 157 in Section 2.

  • 11)The novelty needs further explanation;

Answer:

Innovative further explanation in lines 120 to 133 and 480-495.

  • 12)The reference format is not standard.

Answer:

Reference format has been further modified

Thanks again for your correction.

Round 2

Reviewer 1 Report

My concerns have been properly resolved, but more corrections in English language seem to be required. For example, a period and lower case is required between 'test set' and 'Each blocking' in the sentence 'in the data collection area of the test set Each blocking catetory' on page 4.   

Author Response

For your question, we have made a more comprehensive revision to the English of the article. Modifications have been marked in this chapter.

Thank you again for your corrections and suggestions.

Reviewer 2 Report

1,The abbreviations need full names,  such as  UWB.
2,The format of  references  need to be further modified,  including the abbreviations of journal.
3,DW1000 UWB devices are connected wirelessly. The anchors have not been strictly synchronized. Is TOA method is more reasonable for calculating the position?
4,Can the author provide examples of data formats?What is the main information of an epoch data?

Author Response

1,The abbreviations need full names,  such as  UWB.

Answer:

All abbreviation in the article has been supplemented with the full name. For example, line 17 supplements the full name of UWB. we have made a more comprehensive revision to the English of the article. Modifications have been marked in this chapter.

2,The format of  references  need to be further modified,  including the abbreviations of journal. 

Answer:

All references have been further revised and marked in the article.

3,DW1000 UWB devices are connected wirelessly. The anchors have not been strictly synchronized. Is TOA method is more reasonable for calculating the position?

Answer:

The research in this experiment is the recognition of occlusion categories, and the experimental results have nothing to do with the distance calculation method. However, in actual positioning and ranging applications, if strict clock synchronization cannot be achieved between anchor points, it is more reasonable to adopt the TOA method.

4,Can the author provide examples of data formats?What is the main information of an epoch data?

Answer:

Thank you again for your corrections and suggestions.
